# Approximating the Permanent with Deep Rejection Sampling

**Juha Harviainen**
University of Helsinki
juha.harviainen@helsinki.fi

**Antti Röyskö**
ETH Zürich
aroeyskoe@ethz.ch

**Mikko Koivisto**
University of Helsinki
mikko.koivisto@helsinki.fi

## Abstract

We present a randomized approximation scheme for the permanent of a matrix with nonnegative entries. Our scheme extends a recursive rejection sampling method of Huber and Law (SODA 2008) by replacing the upper bound for the permanent with a linear combination of the subproblem bounds at a moderately large depth of the recursion tree. This method, we call *deep rejection sampling*, is empirically shown to outperform the basic, depth-zero variant, as well as a related method by Kuck et al. (NeurIPS 2019). We analyze the expected running time of the scheme on random $(0, 1)$-matrices where each entry is independently 1 with probability $p$. Our bound is superior to a previous one for $p$ less than $1/5$, matching another bound that was known to hold when every row and column has density exactly $p$.

## 1 Introduction

The permanent of an $n \times n$ matrix $A = (a_{ij})$ is defined as

$$\operatorname{per} A := \sum_{\sigma} a(\sigma), \quad \text{with} \quad a(\sigma) := \prod_{i=1}^{n} a_{i\sigma(i)}, \tag{1}$$

where the sum is over all permutations $\sigma$ on $\{1, 2, \ldots, n\}$. The permanent appears in numerous applications in various domains, including communication theory [31], multi-target tracking [37, 23], permutation tests on truncated data [8], and the dimer covering problem [2]. Bipartite graphs have the well-known property that the permanent of an adjacency matrix of such a graph is equal to the number of its perfect matchings.

Finding the permanent is a notorious computational problem. The fastest known exact algorithms run in time $O(2^n n)$ [29, 15], and presumably no polynomial-time algorithm exists, as the problem is #P-hard [38]. If negative entries are allowed, just deciding the sign of the permanent is equally hard [21]. For matrices with nonnegative entries, a fully polynomial-time randomized approximation scheme, FPRAS, was discovered two decades ago [21], and later the time requirement was lowered to $O(n^7 \log^4 n)$ [3]. However, the high degree of the polynomial and a large constant factor render the scheme infeasible in practice [25]. Other, potentially more practical Godsil–Gutman [16] type estimators obtain high-confidence low-error approximations with $O(c^{n/2})$ evaluations of the determinant of an appropriate random $n \times n$ matrix over the reals [16, 22], complex numbers [22], or quaternions [9], with $c$ equal to 3, 2, and 3/2, respectively. These schemes might be feasible up to around $n = 50$, but clearly not for $n \geq 100$.

35th Conference on Neural Information Processing Systems (NeurIPS 2021).

From a practical viewpoint, however, (asymptotic) worst-case bounds are only of secondary interest: it would suffice that an algorithm outputs an estimate that, with high probability, is guaranteed to be within a small relative error of the exact value—no good upper bound for the running time is required a priori; it is typically satisfactory that the algorithm runs fast on the instances one encounters in practice. The artificial intelligence research community, in particular, has found this paradigm attractive for problems in various domains, including probabilistic inference [1, 7], weighted model counting [12, 11], network reliability [26], and counting linear extensions [35].

For approximating the permanent, this paradigm was recently followed by Kuck et al. [23]. Their AdaPart method is based on rejection sampling of permutations $\sigma$ proportionally to the weight $a(\sigma)$, given in (1). It repeatedly draws a uniformly distributed number between zero and an upper bound $U(A) \geq \operatorname{per} A$ and checks whether the number maps to some permutation $\sigma$. The check is performed iteratively, by sampling one row–column pair at a time, rejecting the trial as soon as the drawn number is known to fall outside the set spanned by the permutations, whose measure is $\operatorname{per} A$. The expected running time of the method is linear in the ratio $U(A)/\operatorname{per} A$, motivating the use of a bound that is as tight as possible. Critically, the bound $U$ is required to "nest" (a technical monotonicity property we detail in Section 2), constraining the design space. This basic strategy of AdaPart stems from earlier methods by Huber and Law [17, 19], which aimed at improved polynomial worst-case running time bounds for dense matrices. The key difference is that AdaPart dynamically chooses the next column to be paired with some row, whereas the method of Huber and Law proceeds in the static increasing order of columns. Thanks to this flexibility, AdaPart can take advantage of a tighter upper bound for the permanent that would not nest with respect to the static recursive partitioning.

In this paper, we present a way to boost the said rejection samplers. Conceptually, the idea is simple: we replace the upper bound $U(A)$ by a linear combination of the bounds for all submatrices that remain after removing the first $d$ columns and the associated any $d$ rows. Here the *depth* $d$ is a parameter specified by the user; the larger its value, the better the bound, but also the larger the time requirement of computing the bound. This can be viewed as an instantiation of a generic method we call *deep rejection sampling* or *DeepAR* (deep acceptance–rejection). Our main observation is that for the permanent the computations can be carried out in time that scales, roughly, as $2^d$, whereas a straightforward approach would scale as $n^d$, being infeasible for all but very small $d$. We demonstrate empirically that "deep bounds", with $d$ around 20, are computationally feasible and can yield orders-of-magnitude savings in running time as compared to the basic depth-zero bounds.

We also study analytically how the parameter $d$ affects the ratio of the upper bound and the permanent. Following a series of previous works [22, 13, 14], we consider random $(0, 1)$-matrices where each entry takes value 1 with probability $p$, independently of the rest. We give a bound that holds with high probability and, when specialized to $d = 0$ and viewed as a function of $n$ and $p$, closely resembles Huber's [17] worst-case bound that holds whenever every row- and column-sum is *exactly* $pn$. We will compare the resulting time complexity bound of our approximation scheme to bounds previously proven for Godsil–Gutman type estimators [22, 13] and some simpler Monte Carlo schemes [27, 14], and argue, in the spirit of Frieze and Jerrum [13, Sec. 6], that ours is currently the fastest practical and "trustworthy" scheme for random matrices.

## 2 Approximate weighted counting with rejection sampling

We begin with a generic method for exact sampling and approximate weighted counting called self-reducible acceptance–rejection. We also review the instantiations of the method to sampling weighted permutations due to Huber and Law [17, 19] and Kuck et al. [23].

### 2.1 Self-reducible rejection sampling

In general terms, we consider the following problem of approximate weighted counting. Given a set $\Omega$, each element $x \in \Omega$ associated with a nonnegative weight $w(x)$, and numbers $\epsilon, \delta > 0$, we wish to compute an $(\epsilon, \delta)$-*approximation* of $w(\Omega) := \sum_{x \in \Omega} w(x)$, that is, a random variable that with probability at least $1 - \delta$ is within a factor of $1 + \epsilon$ of the sum.

The self-reducible acceptance–rejection method [17] solves the problem assuming access to

    (i) a *partition tree* of $\Omega$, that is, a rooted tree where the root is $\Omega$, each leaf is a singleton set, and the children of each node partition the node into two or more nonempty subsets;

(ii) an upper bound that *nests* over the partition tree, that is, a function $u$ that associates each node $S$ a number $u(S)$ that equals $w(x)$ when $S = \{x\}$, and is at least $\sum_{i=1}^{n} u(S_i)$ when the children of $S$ are $S_1, S_2, \ldots, S_n$.

The main idea is to estimate the ratio of $w(\Omega)$ to the upper bound $u(\Omega)$ by drawing uniform random numbers from the range $[0, u(\Omega)]$ and accept a draw if it lands in an interval of length $w(x)$ spanned by some $x \in \Omega$, and reject otherwise. The empirical acceptance rate is known [10] to yield an $(\epsilon, \delta)$-approximation of the ratio as soon as the number of *accepted draws* is at least $\psi(\epsilon, \delta) := 1 + 2.88(1 + \epsilon)\, \epsilon^{-2} \ln(2/\delta)$. The following function implements this scheme by calling a subroutine SAMPLE$(\Omega, u)$, which makes an independent draw, returning 1 if accepted, and 0 otherwise. (Huber's [18] Gamma Bernoulli acceptance scheme, GBAS, reduces the required number of draws to around one third for practical values of $\epsilon$ and $\delta$; see Supplement A.1.)

**Function** ESTIMATE$(\Omega, u, \epsilon, \delta)$

**E1** $k \leftarrow \lceil \psi(\epsilon, \delta) \rceil$, $t \leftarrow 0$, $s \leftarrow 0$

**E2** Repeat $t \leftarrow t + 1$, $s \leftarrow s + $ SAMPLE$(\Omega, u)$ until $s = k$

**E2** Return $u(\Omega) \cdot k/t$

The partition tree and the nesting property are vital for implementing the sampling subroutine, enabling sequential random zooming from the ground set $\Omega$ to a single element of it.

**Function** SAMPLE$(S, u)$

**S1** If $|S| = 1$ then return 1; else partition $S$ into $S_1, S_2, \ldots, S_n$

**S2** $p(i) \leftarrow u(S_i)/u(S)$ for $i = 1, 2, \ldots, n$, $p(0) \leftarrow 1 - \sum_{i=1}^{n} p(i)$

**S3** Draw $i \sim p(i)$

**S4** If $i \geq 1$ then return SAMPLE$(S_i, u)$; else return 0

## 2.2 Application to approximating the permanent

Huber and Law [17, 19] instantiate the method to approximating the permanent of an $n \times n$ matrix $A$ with nonnegative entries by letting $\Omega$ be the set of all permutations on $N := \{1, 2, \ldots, n\}$ and letting the weight function $w$ equal $a$. The recursive partitioning is obtained by simply branching on the row $\sigma^{-1}(j)$ for each column $j = 1, 2, \ldots, n$ in increasing order. The bound $u$ is derived from an appropriate upper bound $U(A) \geq \mathrm{per}\, A$ as follows. Let $S$ be the set of permutations that fix a bijection between rows $I$ and columns $J$ and denote the corresponding column for the row $i$ by $\sigma(i)$. We get the upper bound at $S$ by multiplying the upper bound for the permanent of the remaining matrix by the product of the already picked entries:

$$u(S) := \left( \prod_{i \in I} a_{i\sigma(i)} \right) \cdot U(A_{\bar{I}\bar{J}}). \tag{2}$$

Here $\bar{I}$ and $\bar{J}$ are the complement sets of $I$ and $J$ in relation to $N$, and indexing by subsets specifies a submatrix in an obvious manner. Note that $u(\Omega) = U(A)$ and $u(\{\sigma\}) = a(\sigma)$, provided that we let $U(A) := 1$ when $A$ vanishes, i.e., has zero rows and columns.

Various upper bound for the permanent are known. Let $A_i$ denote the $i$th row vector of $A$, and let $|A_i|$ denote the sum of its entries. For the permanent of a $(0, 1)$-matrix, Minc conjectured [24] and Brègman proved [5] the Minc–Brègman bound

$$U^{\mathrm{MB}}(A) := \prod_{i=1}^{n} \gamma(|A_i|) \ \text{ with } \gamma(0) := 0 \text{ and } \gamma(k) := (k!)^{1/k} \text{ for } k = 1, 2, \ldots.$$

An extension to arbitrary nonnegative weights is credited to Brouwer in Schrijver's work [30] and published in corrected form by Soules [33, cf. Footnote 4]. Letting $a_{i1}^* \geq a_{i2}^* \geq \cdots \geq a_{in}^*$ be the entries of $A_i$ arranged into nonincreasing order, the Brouwer–Schrijver bound is given by

$$U^{\mathrm{BS}}(A) := \prod_{i=1}^{n} \sum_{k=1}^{n} a_{ik}^* \cdot \left[ \gamma(k) - \gamma(k-1) \right].$$

One can verify that for a $(0, 1)$-matrix the bound equals the Minc–Brègman bound.

Since the Minc–Brègman bound does not yield, through (2), a nesting upper bound $u$ over the recursive column-wise partitioning, Huber and Law [19] introduced a somewhat looser upper bound, which has that desired property:

$$U^{\mathrm{HL}}(A) := \prod_{i=1}^{n} \frac{h(|A_i|)}{\mathrm{e}}, \quad h(r) := \begin{cases} r + (\ln r)/2 + \mathrm{e} - 1, & \text{if } r \geq 1, \\ 1 + (\mathrm{e} - 1)r, & \text{otherwise.} \end{cases} \tag{3}$$

We will refer to this as the Huber–Law bound.

In order to employ the tighter Brouwer–Schrijver bound, Kuck et al. [23] replaced the static column-wise partitioning by a dynamic partitioning, where the next column is selected so as to minimize the sum of the bounds of the resulting parts. More formally, if $S$ is the set of permutations that fix a bijection between rows $I$ and columns $J$, then $S$ is partitioned according to a column $j \in \bar{J}$ into sets

$$S_{ij} := \{\sigma \in S : \sigma^{-1}(j) = i\}, \quad i \in \bar{I},$$

such that $\sum_{i \in \bar{I}} u(S_{ij})$ is minimized. Furthermore, if even the smallest sum exceeds the bound $u(S)$, then the partition is refined by replacing some set $S_{ij}$ by its minimizing partition; this is repeated until the nesting condition is met. Kuck et al. report that the initial minimizing partition of $S$ was always sufficient in their experiments; this is vital for computational efficiency.

**Example 1.** It was left open whether the Brouwer–Schrijver bound is guaranteed to nest over the dynamic partitioning. The following matrix $C$ is a counterexample, showing the answer is negative:

$$C := \begin{pmatrix} 1 & 1 & 1 & 1 \\ 0 & 0 & 1 & 1 \\ 1 & 1 & 0 & 0 \\ 1 & 1 & 1 & 1 \end{pmatrix}.$$

Indeed, we have $U^{\mathrm{MB}}(C) = (4!)^{1/2}(2!) = 4\sqrt{6}$, but for any column $j$, the bounds of the three submatrices that remain after removing column $j$ and row $i$ with nonzero entry at $(i, j)$ sum up to $2(3!)^{1/3}(2!)^{1/2}(1!)^{1/1} + (3!)^{2/3}(2!)^{1/2} = (48^{1/3} + 36^{1/3})\sqrt{2} > 2\sqrt{2}(48 \cdot 36)^{1/6} = 4\sqrt{6}$. Here we used the fact that the inequality $(a + b)/2 \geq \sqrt{a \cdot b}$ holds with equality only if $a = b$.

## 3 Deep rejection sampling

This section gives a recipe for boosting self-reducible acceptance–rejection. We formulate our method, DeepAR, first in general terms in Section 3.1, and then instantiate it to the permanent in Section 3.2.

### 3.1 Deep bounds

Consider a fixed partition tree of $\Omega$. Denote by $\mathcal{P}(S)$ the set of children of node $S \subseteq \Omega$. For $d \geq 0$, define the set of *depth-$d$ descendants* of $S$, denoted by $\mathcal{P}_d(S)$, recursively by

$$\mathcal{P}_0(S) := \{S\}, \quad \mathcal{P}_{d+1}(S) := \bigcup_{R \in \mathcal{P}(S)} \mathcal{P}_d(R).$$

We obtain the node set of the partition tree as $\mathcal{P}_* := \bigcup_{h \geq 0} \mathcal{P}_h(\Omega)$.

Suppose $u$ is an upper bound over $\mathcal{P}_*$. Define the *depth-$d$ upper bound* at $S \in \mathcal{P}_h(\Omega)$ as $u_d(S) := u(S)$ if $h > d$, and otherwise as

$$u_d(S) := \sum_{R \in \mathcal{P}_{d-h}(S)} u(R).$$

In words, the depth-$d$ upper bound at node $S$ of the partition tree is obtained by summing up the upper bounds at the nodes in the subtree rooted at $S$ that are at depth $d$ in the whole partition tree. If $u$ nests, then so does $u_d$ and $u_d(\Omega)$ decreases from $u(\Omega)$ to $w(\Omega)$ as $d$ increases; the former is obtained at $d = 0$ and the latter at any sufficiently large $d$ (logarithmic in $|\Omega|$ suffices).

Our idea is to simply replace the basic bound $u$ by the depth-$d$ upper bound in the rejection sampling routine SAMPLE$(S, u)$. This directly increases the acceptance rate by a factor of $u(\Omega)/u_d(\Omega)$, potentially yielding a large computational saving for larger $d$.

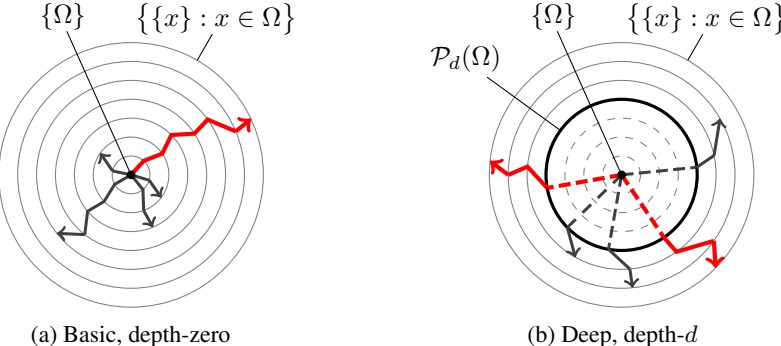

(a) Basic, depth-zero        (b) Deep, depth-$d$

Figure 1: Schematic illustration of basic and deep self-reducible acceptance–rejection. Accepted draws are shown as red thicker arrows, rejected draws as grey arrows.

The main obstacle to efficient implementation of this idea is the complexity of evaluating $u_d(S)$ at a given $S$. For each node $S$ at depth at most $d$, we need to sum up the bounds $u(R)$ at the descendants $R$ of $S$ at depth $d$; straightforward computing is demanding due to the large number of nodes $R$, while precomputing, simultaneously for all $S$, is demanding already due to the number of nodes $S$.

One observation comes to rescue: it suffices that we can *sample* nodes $R$ at depth $d$ in the partition tree proportionally to the respective upper bounds. Put otherwise, we add the following line to the beginning of the SAMPLE function, completing the description of DeepAR:

**S0** If $S = \Omega$ then draw $R \in \mathcal{P}_d(\Omega)$ with probability $u(R)/u_d(\Omega)$, return SAMPLE$(R, u)$

In effect, sampling begins directly at depth $d$, skipping $d$ recursive steps of the original routine (Fig. 1). This allows us to employ, in principle, any algorithm to sample at depth $d$, potentially making use of problem structure other than what is represented by the partition tree. It depends on the problem at hand, more precisely on the upper bound $u$, how efficient algorithms are available. We next see that the upper bounds for the permanent given in Section 2.2 admit a relatively efficient algorithm.

### 3.2 Deep bounds for the permanent

We now implement step S0 for an upper bound $U$ for the permanent. The key property we need is that $U(A)$ factorizes into a product of $n$ terms, the $i$th term only depending on $A_i$, the $i$th row of $A$; all the bounds reviewed in Section 2.2 have this property. We let $\gamma(A_i)$ denote the $i$th term.

Consider a fixed set of columns $J \subseteq N$. Denote $\gamma_i := \gamma(A_{i\bar{J}})$ and $\gamma_I := \prod_{i \in I} \gamma_i$ for short. By summing over all row subsets $I \subseteq N$ of size $d := |J|$, we get

$$\mathrm{per}\,A = \sum_I \mathrm{per}\,A_{IJ}\,\mathrm{per}\,A_{\bar{I}\bar{J}} \leq \overbrace{\sum_I \mathrm{per}\,A_{IJ} \cdot U(A_{\bar{I}\bar{J}})}^{U_d(A)} = \sum_I \mathrm{per}\,A_{IJ} \cdot \gamma_{\bar{I}} = \gamma_N \cdot \sum_I \mathrm{per}\,B_{IJ}\,,$$

where $B = (b_{ij})$ is the rectangular matrix with the index set $N \times J$ and entries $b_{ij} := a_{ij}\,\gamma_i^{-1}$. The sum $\sum_I B_{IJ}$ equals the permanent of $B$ given by $\mathrm{per}\,B := \sum_\tau \prod_{j \in J} b_{\tau(j)j}$, where the sum is over all injections $\tau$ from $J$ to $N$. Thus, in step S0, a row subset $I$ is drawn with probability $\mathrm{per}\,B_{IJ}/\mathrm{per}\,B$. Note that drawing the injection $\tau$ is unnecessary, since it only affects the bound $U(A_{I\bar{J}})$ through the selected rows $I = \tau(J)$.

It remains to show how to generate a random $I$ without explicitly considering all the $\binom{n}{d}$ sets. To this end, we employ an algorithm for the permanent of rectangular matrices due to Björklund et al. [4]. Write $g_i(K) := \mathrm{per}\,B_{IK}$, with $I = \{1, 2, \ldots, i\}$. For $i > 0$ and nonempty $K$, the recurrence

$$g_i(K) = g_{i-1}(K) + \sum_{j \in K} b_{ij} \cdot g_{i-1}(K \setminus \{j\}) \tag{4}$$

enables computing $\mathrm{per}\,B = g_n(J)$ by dynamic programming with $O(2^d dn)$ arithmetic operations:

**Function** $\text{RPER}(B)$

**R1** $g_0(\emptyset) \leftarrow 1$, $g_0(K) \leftarrow 0$ for $\emptyset \subset K \subseteq J$, $i \leftarrow 1$

**R2** For $K \subseteq J$ compute $g_i(K)$ using (4)

**R3** If $i = n$ then return $g$; else $i \leftarrow i + 1$, go to R2

Having stored all the values $g_i(K)$, we generate a random injection $\tau$ with probability proportional to $\prod_{j \in J} b_{\tau(j)j}$ in $O(nd)$ steps by routine stochastic backtracking:

**Function** $\text{DSAMPLE}(g)$

**D1** $i \leftarrow n$, $K \leftarrow J$

**D2** $p(j) \leftarrow b_{ij} \cdot g_{i-1}(K \setminus \{j\})/g_i(K)$ for $j \in K$, $p(0) \leftarrow 1 - \sum_{j \in K} p(j)$

**D3** Draw $j \sim p(j)$

**D4** If $j > 0$ then $\tau(j) \leftarrow i$, $K \leftarrow K \setminus \{j\}$

**D5** If $i = 1$ then return $\tau(J)$; else $i \leftarrow i - 1$, go to D1

To summarize, consider the depth-$d$ variant of the Huber–Law bound, $U_d^{\text{HL}}$. The matrix $B$ can clearly be computed in time $O(n^2)$, with $J := \{1, 2, \ldots, d\}$. Since $U_d^{\text{HL}}$ nests, the number of trials is $O\big(U_d^{\text{HL}}(A)/\text{per}A \cdot \epsilon^{-2} \log \delta^{-1}\big)$, each of which can be implemented in time $O(n^2)$ by simple incremental computing (Supplement A.3). We have shown the following.

**Theorem 1.** *An $(\epsilon, \delta)$-approximation of the permanent of a given $n \times n$ matrix with nonnegative entries can be computed, for any given $0 \le d \le n$, in time $O\big(2^d dn + U_d^{\text{HL}}(A)/\text{per}A \cdot n^2 \epsilon^{-2} \log \delta^{-1}\big)$.*

## 4 An analysis for random matrices

We turn to the question of how well the approximation scheme based on the Huber–Law bound and its deep variant performs on the permanent of a random matrix. Following previous works [22, 14], we focus on $(0, 1)$-matrices of size $n \times n$ where the entries are mutually independent, each entry being 1 with probability $p$, and write "$A \in \mathcal{B}(n, p)$" when $A$ is a matrix in this model. We begin by stating and discussing our main results: a high-confidence upper bound for the ratio of the upper bound to the permanent, and its implication to the total time requirement of the approximation scheme. Then we outline a proof, deferring proofs of several lemmas (marked with a $\star$) to the supplement.

### 4.1 Main results

In our analysis it will be critical to obtain a good *lower bound* for the permanent. To this end, we need the assumption that $p$ does not decrease too fast when $n$ grows.

**Theorem 2.** *Suppose the function $p = p(n)$ satisfies $p^2 n \to \infty$ as $n \to \infty$. Let $\delta > 0$. Then, for all sufficiently large $n$, $A \in \mathcal{B}(n, p)$, and $0 \le d \le n - p^{-1}$, with probability at least $1 - 2\delta$,*

$$\frac{U_d^{\text{HL}}(A)}{\text{per}A} \le \delta^{-1} \big(\pi(n - d)\big)^{-1/2} \big(\mathrm{e}^{2\mathrm{e}-1}(n - d)p_0\big)^{1/(2p_0)}, \tag{5}$$

*where $p_0 := p - n^{-1}\sqrt{2p \ln \delta^{-1}}$.*

**Remark 1.** With $n$ sufficiently large, we could simplify the bound further by replacing $p_0$ with $p$. We present the more involved bound, as it follows more directly from our analysis and because we believe it is within a small factor of a bound that works also for small $n$ with moderate $p$ and $\delta$.

**Remark 2.** In the bound the key parameter is $n - d$ rather than $n$, and the effect of $d$ is linear in the base of the exponential bound. It remains open whether there exists a class of matrices where the ratio grows as $c^n$, $c > 1$, for the depth-zero bound, and as $c^{(1-\rho)n}$ when the depth is $\rho n$.

The following time complexity result is an immediate corollary to Theorems 1 and 2.

**Theorem 3.** *For any fixed $\alpha > 1/2$ and $\epsilon, \delta, p > 0$, an $(\epsilon, \delta)$-approximation of the permanent of a random matrix $A \in \mathcal{B}(n, p)$ can be, with high probability, computed in expected time $O(n^{1.5+\alpha/p})$.*

Huber's [17] closely related rejection sampling method, based on a slightly different nesting upper bound for the permanent, is known to run in expected time $O(n^{1.5+0.5/\rho})$ for *all* $(0, 1)$-matrices where every row and column has *exactly* $\rho n$ ones. Thus, our result extends Huber's also to matrices where row- and column-sums deviate from the expected value, some more and some less. On the other hand, Huber and Law's [19] related scheme runs in expected time $O(n^{1.5+0.5/(2\rho-1)})$ for all $(0, 1)$-matrices where every row and column sum is *at least* $\rho n$ and $\rho > 0.5$; this bound, adopted also to the AdaPart scheme [23, Prop. 1], is larger than the previous bound for all $\rho < 1$.

We have also mentioned other Monte Carlo estimators that do not use rejection sampling. Their running time depends crucially on the variance of the estimator $X$, or, more precisely, on the so-called critical ratio $\mathbf{E}[X^2]/\mathbf{E}[X]^2$. For random matrices, high-confidence upper bounds for this ratio have been proven for a determinant based estimator [22, 13] and a simpler sequential estimator [14], yielding approximation schemes that run in time $O(n\,t_1(n)\,\omega(n))$ and $O(t_2(n)\,\omega(n))$, respectively; here $t_1(n) = O(n^3)$ the time complexity of computing the determinant, $\omega(n)$ is any function tending to infinity as $n \to \infty$, and $t_2(n)$ equals $n^2$, $n^2 \ln n$, or $n^{1/p-1}$, depending on whether $p$ is greater than, equal to, or less than $1/3$, respectively. We observe that the latter bound is worse than ours for $p < 1/5$ and better for $p > 1/5$, while the former bound, being $O(n^4)$, is the best for $p \leq 1/5$.

**Remark 3.** The determinant based estimator has an important weakness: No efficient way is known for giving a good upper bound for the critical ratio for a given input matrix. Thus, either one has to resort to a known pessimistic (exponential) worst-case bound, or terminate the computations after some time with uncertainty about whether the estimate enjoys the accuracy guarantees. The latter variant is not "trustworthy." For further discussion and a precise formalization of this notion, we refer to Frieze and Jerrum [13, Sec. 6]. They also point out that another estimator based on the Broder chain [6, 20] is trustworthy and polynomial-time for random matrices—Huber [17, Sect. 3.1], however, shows that that scheme is slower than the rejection sampling based scheme.

## 4.2   Proof of Theorem 2: outline

It is easy to compute the *expected value* of the permanent. It is also relatively straightforward to compute a good upper bound for the *expected value* of the Huber–Law bound (cf. Lemma 6 below). However, the ratio of these expected values only gives us a hint about the upper bound of interest; in particular, the ratio of expected values can be much smaller than the expected value of the ratio. To overcome this challenge, we use the insight of Frieze and Jerrum [13] that "[the permanent] depends strongly on the number of 1s in the matrix, but only rather weakly on their disposition." Accordingly, our strategy is to work conditionally on the number of 1s, and only at the end get rid of the condition.

For brevity, write $V$ for $\mathrm{per}\,A$ and $U$ for $U_d^{\mathrm{HL}}(A)$. Let $M$ be the number of 1s in $A$. By a Chernoff bound, the binomial variable $M$ is below $m_0 := n^2 p - a$ with probability at most $\exp\{-a^2 n^{-2} p^{-1}/2\}$, which is at most $\delta$ if we put $a := n\sqrt{2p \ln \delta^{-1}}$. From now on, we will assume that $M = m$, with $m \geq m_0$, and give an upper bound for $U/V$ that fails with probability at most $\delta$; the overall success probability is thus at least $1 - 2\delta$. The upper bound will be a decreasing function of $m$, and substituting the lower bound $m_0$ for $m$ will yield the bound in the theorem.

For what follows we need that $n^3 m^{-2}$ tends to zero as $n$ grows. This holds under our assumptions, since $mn^{-3/2} \geq n^{1/2}p - n^{-3/2}a$, where the first term tends to $\infty$ and the second to 0 as $n \to \infty$.

We begin by considering the ratio of the expected values, $\mathbf{E}[U \mid M = m]/\mathbf{E}[V \mid M = m]$. For the denominator, we use a result of Frieze and Jerrum [13]:

**Lemma 4** ([13, Eq. (4)]). *We have* $\mathbf{E}[V \mid M = m] = n!\left(\frac{m}{n^2}\right)^n \exp\left\{-\frac{n^2}{2m} + \frac{1}{2} - O\left(\frac{n^3}{m^2}\right)\right\}$.

For the numerator, we derive an upper bound by reverting back to the independence model, i.e., without conditioning on the number of 1s, for then the calculations are simpler. First, we apply Markov's inequality to relate the two quantities:

**Lemma 5** ($\star$). *If* $p = mn^{-2}$, *then* $\mathbf{E}[U \mid M = m] \leq 2\,\mathbf{E}[U]$.

Then we use the concavity of the function $h$ in Eq. (3) and Jensen's inequality.

**Lemma 6** ($\star$). *We have* $\mathbf{E}[U] \leq \binom{n}{d}d!\,p^d\,\mathrm{e}^{d-n}\,h\big(p(n-d)\big)^{n-d}$.

Combining the above three lemmas and simplifying yields the following:

**Lemma 7** ($\star$). *We have*

$$\frac{\mathbf{E}[U \mid M = m]}{\mathbf{E}[V \mid M = m]} \leq e^{O(n^3 m^{-2})} \left(\pi e(n-d)/2\right)^{-1/2} \left(e^{2e-1} p_*(n-d)\right)^{1/(2p_*)},$$

*where $p_* := mn^{-2}$. Furthermore, the upper bound decreases as a function of $m$.*

Next we show that with high probability $U$ is not much larger and $V$ not much smaller than expected. For the former, direct application of Markov's inequality suffices. For the latter we, again, resort to a result of Frieze and Jerrum, which enables application of Chebyshev's inequality:

**Lemma 8** ([13, Theorem 4]). *We have $\mathbf{E}[V^2 \mid M = m] \big/ \mathbf{E}[V \mid M = m]^2 = 1 + O(n^3 m^{-2})$.*

This yields the following upper and lower bounds:

**Lemma 9** ($\star$). *Conditionally on $M = m$, we have with probability at least $1 - \alpha - \beta^{-2} O(n^3 m^{-2})$,*

$$U \leq \alpha^{-1} \mathbf{E}[U \mid M = m] \quad and \quad V \geq (1 - \beta) \mathbf{E}[V \mid M = m].$$

We now choose $\alpha$ and $\beta$ such that the failure probability is at most $\delta$. Since $n^3 m^{-2}$ tends to zero as $n$ grows, we could set $\beta$ to an arbitrarily small positive value, and $\alpha$ to any value smaller than $\delta$. In particular, we can choose $\alpha$ and $\beta$ such that $\alpha^{-1}(1-\beta)^{-1} \exp\{O(n^3 m^{-2})\}(e/2)^{-1/2} \leq \delta^{-1}$ for sufficiently large $n$. Combining Lemmas 7 and 9 and substituting $m_0$ to $m$ yields the bound (5).

## 5 Empirical results

We report on an empirical performance study of DeepAR for exact sampling of weighted permutations and for approximating the permanent. We include our C++ code and other materials in a supplement.

### 5.1 Tested rejection samplers and approximation schemes

Both the Huber–Law bound and the Brouwer–Schrijver bound are computed by applying a function on each row and taking the product of the results, and thus our dynamic programming method works with them. We have implemented the following instantiations of DeepAR:

   HL-$d$: the scheme due to Huber using the depth-$d$ Huber–Law bound.

   ADAPART-$d$: the AdaPart scheme using the depth-$d$ Brouwer–Schrijver bound.

The time requirement per trial is $O(n^2)$ for HL-$d$ and $O(n^3)$ for ADAPART-$d$, provided that there is always a column on which the Brouwer–Schrijver bound is nesting (Supplement A.3).

We ran the schemes with depths $d \in \{0, 20\}$ and with and without preprocessing of the input matrix. The preprocessing is similar to one by Huber and Law [19]: First, we ensure that every entry belongs to at least one permutation of nonzero weight [36]. Then we apply Sinkhorn balancing $n^2$ times [34] to obtain a nearly doubly stochastic matrix (Supplement A.2). Finally, we divide each row vector by its largest entry. This preprocessing, we abbreviate as DS, takes $O(n^4)$ time. When the input matrix was preprocessed in this way, we add the suffix "-DS" to the names HL-$d$ and ADAPART-$d$.

For comparison, we also ran Godsil–Gutman type schemes (our implementations) and the original authors' implementation of AdaPart. Our main finding is that Godsil–Gutman type schemes are not competitive and that our implementation, ADAPART-0, is typically one to two orders of magnitude faster than the original one; see Supplement C for more detailed report on these experiments.

### 5.2 Estimates of expected running times

We estimated the *expected running time (ERT)* of the schemes for producing a $(0.1, 0.05)$-approximation of the permanent. Using GBAS [18], this yields a requirement of $388$ accepted samples. To save the total computing time of the study, we estimate the ERT based on $65$ accepted samples. By the argument of Section 2.1, our estimate of the ERT has a relative error at most $50 \%$ with probability at least $95 \%$ (if we ignore the variation in the running times of rejected trials).

We considered three classes of random matrices: in *Uniform* the entries are independent and distributed uniformly in $[0, 1]$; *Block Diagonal* consists of block diagonal matrices whose diagonal

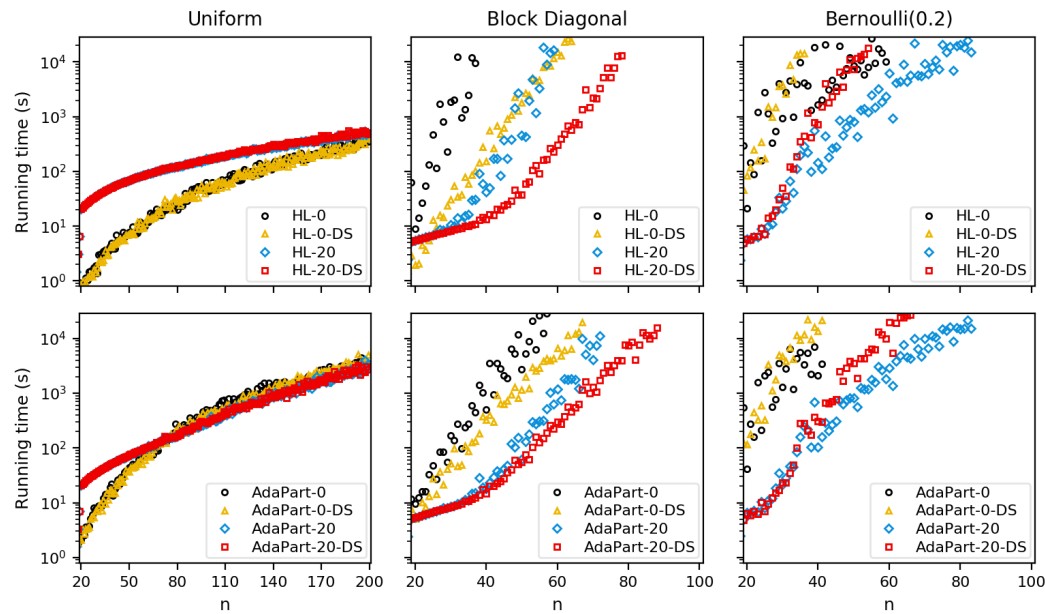

Figure 2: Estimates of expected running times on random instances.

elements are independent $5 \times 5$ matrices from Uniform (the last element may be smaller); in *Bernoulli(p)*, the entries are independent taking value 1 with probability $p$, and 0 otherwise. In Figure 2, with HL-$d$ on the upper row and ADAPART-$d$ on the bottom row, we observe that on Uniform, deep bounds do not pay off and that the Huber–Law scheme is an order-of-magnitude faster than AdaPart. On Block Diagonal, deep bounds bring a significant, two-orders-of-magnitude boost and AdaPart is slightly faster than Huber–Law; furthermore, DS brings a substantial additional advantage for larger instances (visible as a smaller slope of the log-time plot). Bernoulli($p$) again shows the superiority of deep bounds, but also a case where DS is harmful.

We also considered (non-random) benchmark instances. Five of these are from the Network Repository [28] (http://networkrepository.com, licensed under CC BY-SA), the same as included by Kuck et al. [23]. In addition, we included *Staircase-n* instances, which are $(0, 1)$-matrices $A = (a_{ij})$ of size $n \times n$ such that $a_{ij} = 1$ if and only if $i + j \leq n + 2$. Soules [32] mentions these as examples where the ratio of the upper bound and the permanent is particularly large. We observe (Table 1) that on four of the five instances from the repository, the preprocessing has relatively little effect, deep bounds yield a speedup by one to two orders of magnitude, and the configurations of AdaPart and the Huber–Law scheme are about equally fast. The exception is cage5, on which the Huber–Law bound breaks down without DS; a closer inspection reveals that is due to row-sums that are less than 1. On the Staircase instances, deep bounds yield a dramatic speedup and DS makes a clear difference.

Table 1: Estimates of expected running times (in seconds) on benchmark instances.

| | | ADAPART-$d$ | | ADAPART-$d$-DS | | HL-$d$ | | HL-$d$-DS | |
|---|---|---|---|---|---|---|---|---|---|
| Instance | $n$ | $d = 0$ | $d = 20$ | $d = 0$ | $d = 20$ | $d = 0$ | $d = 20$ | $d = 0$ | $d = 20$ |
| ENZYMES-g192 | 31 | $2 \cdot 10^2$ | $\mathbf{1 \cdot 10^1}$ | $8 \cdot 10^2$ | $\mathbf{1 \cdot 10^1}$ | $1 \cdot 10^2$ | $\mathbf{1 \cdot 10^1}$ | $6 \cdot 10^2$ | $2 \cdot 10^1$ |
| ENZYMES-g230 | 32 | $3 \cdot 10^2$ | $2 \cdot 10^1$ | $1 \cdot 10^3$ | $\mathbf{1 \cdot 10^1}$ | $2 \cdot 10^2$ | $\mathbf{1 \cdot 10^1}$ | $1 \cdot 10^3$ | $2 \cdot 10^1$ |
| ENZYMES-g479 | 28 | $3 \cdot 10^2$ | $8 \cdot 10^0$ | $1 \cdot 10^3$ | $8 \cdot 10^0$ | $1 \cdot 10^2$ | $\mathbf{7 \cdot 10^0}$ | $7 \cdot 10^2$ | $9 \cdot 10^0$ |
| cage5 | 37 | $2 \cdot 10^3$ | $2 \cdot 10^1$ | $1 \cdot 10^3$ | $\mathbf{1 \cdot 10^1}$ | $> 10^4$ | $5 \cdot 10^3$ | $9 \cdot 10^2$ | $2 \cdot 10^1$ |
| bcspwr01 | 39 | $6 \cdot 10^2$ | $\mathbf{1 \cdot 10^1}$ | $4 \cdot 10^3$ | $2 \cdot 10^1$ | $3 \cdot 10^2$ | $\mathbf{1 \cdot 10^1}$ | $2 \cdot 10^3$ | $2 \cdot 10^1$ |
| Staircase-30 | 30 | $> 10^4$ | $2 \cdot 10^1$ | $9 \cdot 10^3$ | $8 \cdot 10^0$ | $> 10^4$ | $2 \cdot 10^1$ | $3 \cdot 10^3$ | $\mathbf{7 \cdot 10^0}$ |
| Staircase-45 | 45 | $> 10^4$ | $> 10^4$ | $> 10^4$ | $2 \cdot 10^3$ | $> 10^4$ | $> 10^4$ | $> 10^4$ | $\mathbf{8 \cdot 10^2}$ |

# 6 Concluding remarks

DeepAR (deep acceptance–rejection) boosts recursive rejection sampling by replacing the basic upper bound by a deep variant. While efficient implementations of DeepAR to concrete problems remain to be discovered case by case, we demonstrated the prospects of the method on the permanent of matrices with nonnegative entries, an extensively studied hard problem. DeepAR enables smooth trading of precomputation for acceptance rate in the rejection sampling, and in our empirical study showed expedition by up to *several orders of magnitude*, as compared to the recent AdaPart method [23] (the original authors' implementation). The speedup varies depending on the size of the matrix and the tightness of the upper bound, and can be explained by three factors. A factor of 10–100 is due to implementation details, including both the different programming language and our more streamlined evaluation of the bounds for submatrices. Another factor of 2–1000 is due to our deep bounds. The third factor is due to the preprocessing of the matrix towards doubly-stochasticity (DS), which yields large savings on some hard instances, being mildy harmful for some others. A topic for further research is automatic selection of the best configuration (the permanent bound, depth, DS) on per-instance basis. There are also intriguing analytic questions (cf. Remark 2).

## Acknowledgements

Research partially supported by the Academy of Finland, Grant 316771.

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
