# OpenReview forum: "Approximating the Permanent with Deep Rejection Sampling"
_NeurIPS.cc/2021/Conference — NeurIPS 2021 Poster_

### Official Review · Reviewer_8km5 · 2021-07-10

**Rating:** 5
**Confidence:** 4

**Summary:**

The paper gives a new algorithm to approximate permanents of non-negative matrices. The method builds upon the well-known rejection sampling method by Huber-Law (SODA'08). The algorithm is experimentally shown to perform well, and in particular better than the related result by Kuck et al. (NeurIPS'19). Finally, a mathematical analysis of the performance of the algorithm on a family of random matrices is given, where it is shown to run in polynomial time to yield arbitrarily good precision.

**Limitations And Societal Impact:**

-

**Main Review:**

The paper studies approximating weighted counting problems such as the permanent using an idea based on self-reducible rejection sampling by Huber-Law. This is a well-studied heuristic for approximating permanents. Roughly speaking, in this method we have a certain upper bound on the permanent per(A)<=U(A) and we keep sampling permutations (according to a specially crafted distribution resulting from self-reducibility) until we have seen at least K "yes" samples. Finally the estimate is given by U(M)*K/T, where T is the total number of samples drawn. It is not hard to see that the running time of this method depends on the quality of the upper bound U(M) and what this paper achieves is replacing this bound by a tighter one. The new bound U_d(M) obtained in this paper is a generalization of the Huber-Law bound, based on a kind of d-level deep look-ahead in the tree that is used in the sampling process. The regular Huber-Law bound U(M) is U_1(M).

The presentation of the paper's results is clear and the paper is generally easy to read. The basic idea of self-reducible rejection sampling is explained with enough details before moving on with the presentation of this paper's contribution: deep rejection sampling.

The first result in the paper (Theorem 1) asserts that in time roughly O(2^d*dn + U_d(A)/per(A)*poly(n, 1/eps)) one can compute an eps-approximation of the permanent per(A) of an nxn matrix with >=0 entries (with logarithmic dependency on the error probability). In general, it is not simple to give reasonable bounds on U_d(A), hence the authors provide an analysis of U_d(A) for random Bernoulli matrices. They show (Theorem 2) that in this case the algorithm runs in polynomial time, and beats known results in the regime where p (the probability of an entry being 1) is <1/5.

Most of the proofs are delegated to the supplementary materials, yet the provided overviews are valuable and allow to shape some intuitions.

Authors provide an empirical evaluation of their results comparing their algorithm with the one by Kuck et al. When presenting the results it would be much cleaner if both methods were plotted on a single diagram instead of having separate diagrams per method -- this way it is not clear how do they compare (which seems to be the main point of this experiment).

When assessing the paper I had the following doubt: shall I consider it as A) a theoretical paper with the main contribution being the mathematical theorems, or B) an applications-focused paper, giving a practical heuristic for counting problems. When treated as A) I have concerns that the contribution of this paper over previous work (Huber-Law and then Kuck et al.) is not that significant and hence I would not place it among top papers. When treated as B), I felt that this paper does not provide enough discussion on *concrete* possible applications in learning and what the concrete impact of this new method could be.

**Time Spent Reviewing:**

6

---

> ### Author Response · Authors · 2021-08-09
> **Response to Reviewer 8km5**
>
> Thank you for the preliminary review. We’d like to use this opportunity to correct some inaccuracies or misconceptions in it.
>
> First, we think “is a generalization of the Huber-Law bound” overlooks the fact that our construction works for any bound that is a product of row bounds, in particular, it works for the Schrijver–Soules bound employed in the AdaPart paper.
>
> Second, we disagree with the viewpoint that “it would be much cleaner if both methods were plotted on a single diagram instead of having separate diagrams per method”; by “both” the Reviewer refers to (1) our novel method and to (2) the method of Kuck et al. Namely, we *do* plot both methods on a single diagram: the bottom row of Figure 2. Note that AdaPart-0 is (our just much faster implementation of) Kuck’s et al. AdaPart, while the remaining 3 curves are for variants of our scheme. It seems like the Reviewer has not realized that our scheme (DeepAR) works for both the Huber–Law bound (HL) and the Schrijver–Soules bound (SS), and that for this reason we compare the proposed novel scheme using both, resulting in two rows in Figure 2. Comparing our schemes to the depth-0 variants is of primary interest; comparing HL against SS is of secondary interest. That said, we will consider adding diagrams that enable easier comparison of HL versus SS to the supplement.
>
> Third, we see no need to classify a paper as a theory (A) or applications (B) paper. The NeurIPS 2021 call for papers doesn’t seem to support such a viewpoint, and indeed, NeurIPS has a long and strong tradition in publishing *methods development* papers, where the contributions are not only in the proved theorems but also in experiments demonstrating the significance of the conceptual and technical contributions in practice, e.g., computational efficiency or accuracy in learning from data (often focusing on synthetic and benchmark instances or datasets). Given that we give a conceptual contribution (deep AR), technical contribution (efficient computation using dynamic programming), analytic contribution (non-trivial analysis of random input matrices), and an empirical contribution (efficient implementations and demonstrations of running time savings by several orders of magnitude), we’d like to ask the Reviewer to be more specific about the possible concerns of novelty or significance.

---

> > ### Comment · Reviewer_8km5 · 2021-08-16
> > **Response**
> >
> > Thanks for clarifying the issue with Figures. That was my initial understanding that in Figure (2) your method and Kuck et al. are compared, yet the significant speedup is nowhere to be seen in this figure (for small instances AdaPart-0 seems to win, and for larger one it does equally well), that's why I got confused.
> >
> > Regarding the issue with (A) vs (B). I'm of the opinion that the technical contribution (with respect to previous work) of this paper is way too low for it to be accepted to any strong theory conference such as FOCS/STOC/SODA, on the other hand, when it comes to practical applications this paper does not seem to offer concrete ideas either, only gives some mandatory experiments (which are fine, and well prepared, they are just not interesting from the practical angle). If these are the kind of papers that are desired for NeurIPS, then fine.
> >
> > I will leave my score unchanged.

---

> > > ### Author Response · Authors · 2021-08-18
> > > **Response**
> > >
> > > Thanks for the response. We’d like to further emphasize that AdaPart-0 wins *nowhere* in the results reported in Figure 2: On Uniform, each variant of HL is everywhere better than the respective variant of AdaPart, and AdaPart-0-DS is slightly better than AdaPart-0 on the smallest instances. On Block Diagonal, on matrices of size 60 x 60, AdaPart-0 is about 100 times slower than AdaPart-20. So, the interpretation that “the significant speedup is nowhere” is simply incorrect.
> > >
> > > The additional remarks regarding the A/B classification enforce our understanding that the Reviewer holds a rather different opinion compared to us about the scope of NeurIPS. We are wondering how the Reviewer would classify the NeurIPS paper by Kuck et al., the closest reference to ours.

---

### Official Review · Reviewer_EvRy · 2021-07-16

**Rating:** 9
**Confidence:** 3

**Summary:**

The authors addressed the efficient approximation of permanent of a matrix with nonnegative entries. The proposed scheme is based on the recursive rejection sampling method by Huber 2008, but the upper bound on the permanent is based on the linear combination of the subproblems with the deep recursion tree. The expected running time is both theoretically and experimentally analyzed.

**Ethical Concerns:**

None.

**Limitations And Societal Impact:**

Yes.

**Main Review:**

I think the research is quite interesting overall. The algorithmic idea (previous rejection sampling-based method with deep bounds), related work, algorithm, results, proof sketch are quite well presented. Both theoretical and numerical performance improvement is quite significant. Due to the importance of the problem of permanent computation, I believe that this result will be very valuable.

The only comment I can make is that Theorem 2 is not directly compared with the complexity result in [23]. I think it is fair to make an accurate comparison including both improvements and limitations.

**Time Spent Reviewing:**

10

---

> ### Author Response · Authors · 2021-08-09
> **Response to Reviewer EvRy**
>
> Thank you for the positive review. The complexity result in Proposition 1 in ref. 23 (the AdaPart paper) is, in essence, a re-statement of the result by Huber and Law ([19], Law’s PhD thesis) and, unfortunately, does not explicitly state the following key assumptions: (i) the density parameter gamma must be greater than 0.5 (a very strong condition!); (ii) the input matrix must have undergone certain preprocessing (not mentioned in the AdaPart paper). Moreover, the statement omits the time complexity of running AdaPart until it fails, after which the algorithm resorts to the one by Huber and Law (whence the complexity result) – the omitted complexity of AdaPart may dominate the total complexity, rendering Prop. 1 incorrect strictly speaking. With these issues in mind, we thought that our choice to discuss the other previous theoretical results in Section 4.1 was better use of the limited space. That said, we agree that it would be good to briefly compare our complexity results to that of Huber and Law, with a note about Prop. 1 in ref. 23, in the final version of the paper. Note that we already do refer to the result in lines 42–44, by “This basic strategy of AdaPart stems from earlier methods by Huber and Law [17, 19], which aimed at improved polynomial worst-case running time bounds for dense matrices.”

---

### Official Review · Reviewer_3uKV · 2021-07-20

**Rating:** 6
**Confidence:** 2

**Summary:**

The paper proposes a randomized method for estimating the permanent positive matrix. It uses a recursive rejection sampling method but with a more fine-grained upper bound and provides an efficient way to compute it. They provide theoretical analysis on expected running time for random matrices, where the bound is superior for certain regimes of p, and provide compelling empirical experiments to show practical improvement over prior methods.

**Limitations And Societal Impact:**

Yes, in sections 4 and 5.

**Main Review:**

I think the paper is well-structured, has nice theoretical results with competitive practical performance. It is a good step toward understanding  efficient permanent approximation algorithms. I only have three small concerns regarding the work:

1) I am not sure NeurIPS is the best venue for this work, as it is quite theoretical, and I am unsure about how significant it is a problem of estimating permanent for the general machine learning community.

2) The framework seems to rely heavily on the recursive rejection sampling method in prior work, and the main novelty in idea is the deep bound depending on depth instead of a uniform bound to increase acceptance probability for better efficiency.

3) Also, the runtime has an infavorable 2^d factors, which doesn't seem very scalable.

I would be happy to increase my score if authors offer compelling arguments against my concerns.

**Time Spent Reviewing:**

2.5

---

> ### Author Response · Authors · 2021-08-09
> **Response to Reviewer 3uKV**
>
> Thank you for the preliminary review and the raised small concerns.
> 1. We believe NeurIPS is the best publication venue for our paper. The main reason is that the key reference work (introducing AdaPart) was published in NeurIPS 2019; the main contribution of that work was similarly “theoretical”, i.e. methods development, the application to multi-target tracking serving rather as a demonstration. Note that NeurIPS has a long and strong tradition in publishing research on sampling based methods (various variants of Monte Carlo). The permanent not only has notable applications in computational statistics and machine learning, but also is among the most fundamental benchmark problems for the development of novel computational schemes.
> 2. Yes, our main innovation is the deep bound. As far as we know, the conceptual idea is novel, the relatively efficient implementation using dynamic programming is original, and the significance in terms of improvement upon previous work in practice is rather dramatic (faster by several orders of magnitude). We do not see a reason for concern.
> 3. It is true that 2^d is not favorable for large d. Still, it is feasible for moderately large d, say d = 20, which is sufficient for obtaining significant gains in efficiency even for large matrices, as we show in the paper. It is worth noting that, given the hardness of the problem and extensive previous research (summarized in the second paragraph of Section 1), it may not be realistic to hope for anything “very scalable.”

---

### Official Review · Reviewer_kjA5 · 2021-07-21

**Rating:** 6
**Confidence:** 3

**Summary:**

Computing the matrix permanent is known as a #P-complete problem and no exact polynomial-time algorithm is known. Several works developed a randomized algorithm for approximating the permanent of a non-negative matrix based on recursive acceptance-rejection (AR) sampling and various upper bounds of permanent. This paper adopts a deep AR sampling scheme that skips some recursive steps. Then, the authors additionally propose an efficient way for computing skipped steps based on dynamic programming of permanent. Empirical results demonstrate order-of-magnitude savings of computational time under random synthetic and real-world datasets.

**Ethics Review Area:**

["I don’t know"]

**Limitations And Societal Impact:**

Described above

**Main Review:**

The main contribution of this work is boosting recursive sampling by skipping intermediate steps. Although the method is simple, the authors additionally propose an efficient way of computing the upper bound at depth d. This is a novel approach that utilizes the key properties of permanent. Furthermore, experimental results show a huge gap of computational savings under real-world datasets.
Some questions are the following:
- In Theorem 1, it is clear that the running time depends on the depth d, but unclear how the depth affects an approximation quality. It would be great if the approximation quality is provided in terms of the choice of depth.
- The depth can be specified by the user, but the algorithm varies on the choice of the depth. Is it possible to theoretically/empirically predict the optimal value of depth? Simply, does the optimal depth that minimizes the running time of the algorithm provided in Theorem 1?
- In section 5, what is the approximation quality (or accuracy) of each benchmark algorithm?  It would be great to provide accuracy comparison.
- Additionally, it would be much better if the paper is polished more so that it can be much easier to understand the proposed algorithm and related works.

Minor questions:
- In equation (2), does the summation over all permutations \sigma of S drop?
- Is the \gamma (in line 114-115) well-defined for k=0?
- The toy example provided in section 4 seems to be related to an adjacency matrix of Erdos-Renyi graph. If so, the proposed algorithm can estimate the number of perfect matchings of a given graph. Some readers may welcome it if the paper provides a corollary of perfect matching counting algorithm.


**Time Spent Reviewing:**

7

---

> ### Author Response · Authors · 2021-08-09
> **Response to Reviewer kjA5**
>
> Thank you for the preliminary review and questions.
>
> We’d like to use this opportunity to correct two lines in the summary part of the review. First, we think that “adopts a deep AR sampling scheme” partly misses the key point that our paper is the first to introduce the idea of “deep AR”: we introduce/propose/present rather than adopt. Second, we think “order-of-magnitude savings” gives a biased summary of the empirical results, since our method is faster than the previous state of the art by *several orders of magnitude.*
>
> Our responses to the four main questions:
>  -	The parameter d does not affect the approximation quality. The approximation quality is quantified by epsilon and delta, which are independent of the other input parameters. Here we follow the standard practice in reporting efficiency results for randomized approximation schemes. We see no advantage of making epsilon and delta dependent on the depth d.
> -	Yes, depth d is optimal if it minimizes the (expected) running time. Note that constant factors, hidden by the O-notation, also matter in practice. Another complication is that the value of the permanent is, of course, unknown. We believe that despite these complications, one can develop a scheme that automatically and efficiently finds a nearly optimal d, but we have left this development for future work.
> -	Like we state in line 301, we compare the schemes in a setting where each guarantees an (0.1, 0.05)-approximation of the permanent; we recall the definition of (epsilon, delta)-approximation in lines 74–75. Because we only compare schemes that have provable and controllable accuracy, we see little added value in comparing the produced estimates against the actual permanent (which we can compute exactly only for small or special matrices). If, instead, one tested methods that do not enjoy accuracy guarantees (e.g., typical instantiations of MCMC in machine learning), then comparison against the known ground truth would be essential.
> -	Thank you for suggestions to improve the clarity of the presentation.
>
> Our answers to the minor questions:
> -	There is no summation over sigma. Note that S is a fixed set of permutations, in which every permutation is identical when (as a mapping from the rows to the columns) restricted to the row set I. Because this part is identical, we may refer to it by the same symbol sigma. (We admit that is a somewhat technical part, which we can explain with more words in the final version of the paper.)
> -	We have to add that gamma(0) := 0. Thank you.
> -	As far as we know, Erdős–Rényi graphs exclusively refer to *random* graphs. So, our particular, small, non-random example has no immediate relation to it. Nevertheless, it is true and *very* well known that the permanent of an adjacency matrix of a *bipartite graph* G equals the number of perfect matchings of G. Because this relation is so well known, we think that adding a corollary would add very little to the paper. But we will consider recalling this relation in the opening paragraph of Section 1.

---

### Decision · Program_Chairs · 2021-09-27

**Decision:**

Accept (Poster)

**Comment:**

This paper gives a generalization of an algorithm of Huber and Law for approximating the permanent of a non-negative matrix (an important and basic counting/sampling problem). The idea is to combine the rejection sampling method of Huber and Law with a deep (parameterized by some depth parameter d) "look-ahead" that uses a linear combination of sub-problems at depths <= d in the recursion tree as an upper bound on the permanent. This theoretically strengthens the guarantees of Huber and Law (and beats them on the "toy" example of random Bernoulli matrices. In experiments it beats the guarantees of a recent work of Kuck et. al. in approximating the permanent.

There were concerns that the theoretical contributions of the paper are not strong enough on top of prior work. However, the relatively clean and simple look-ahead method (deep AR) developed in this paper does give a significant advantage in terms of practical speed-ups. The deep AR heuristic, as a result, appears an appealing avenue to explore further. We recommend acceptance.